# Subchronic Exposure to Polystyrene Microplastic Differently Affects Redox Balance in the Anterior and Posterior Intestine of *Sparus aurata*

**DOI:** 10.3390/ani13040606

**Published:** 2023-02-09

**Authors:** Filomena Del Piano, Adriano Lama, Anna Monnolo, Claudio Pirozzi, Giovanni Piccolo, Simone Vozzo, Davide De Biase, Lorenzo Riccio, Giovanna Fusco, Raffaelina Mercogliano, Rosaria Meli, Maria Carmela Ferrante

**Affiliations:** 1Department of Veterinary Medicine and Animal Productions, Via Delpino 1, 80137 Naples, Italy; 2Department of Pharmacy, University of Naples Federico II, Via D. Montesano 49, 80131 Naples, Italy; 3Department of Pharmacy, University of Salerno, Via Giovanni Paolo II 132, 84084 Fisciano, Italy; 4Zooprophylactic Institute of Southern Italy, Via Salute 2, 80055 Portici, Italy

**Keywords:** polystyrene microplastic, subchronic exposure, anterior and posterior intestine, oxidative stress, antioxidant defence

## Abstract

**Simple Summary:**

Plastics are among the most widely distributed pollutants in the sea, where they break down into microplastic particles that undergo bioaccumulation and biomagnification processes along the trophic chain. Such particles can cause toxic effects in aquatic organisms, such as cytotoxicity, inflammation, and oxidative stress. Gilthead seabream (*Sparus aurata*) is one of the most spread species in Mediterranean aquaculture fish. It is a species at the top of the trophic chain and is considered a good sentinel species for toxicological and bioaccumulation studies. This study aimed to investigate the adverse effects of the oral exposure to polystyrene microplastics on intestinal redox balance. We demonstrated that microplastics ingestion increases the intestinal oxidative and nitrosative stress and impairs the antioxidant defense system. This evidence confirms that the gut is a target organ for the toxic effects of ingested microplastics in fish. The observed impairment may damage the organ function, leading to the alteration of fish health status.

**Abstract:**

Microplastics (MPs) are pollutants widely distributed in aquatic ecosystems. MPs are introduced mainly by ingestion acting locally or in organs far from the gastroenteric tract. MPs-induced health consequences for fish species still need to be fully understood. We aimed to investigate the effects of the subchronic oral exposure to polystyrene microplastics (PS-MPs) (1–20 μm) in the gilthead seabreams (*Sparus aurata*) used as the experimental model. We studied the detrimental impact of PS-MPs (25 and 250 mg/kg b.w./day) on the redox balance and antioxidant status in the intestine using histological analysis and molecular techniques. The research goal was to examine the anterior (AI) and posterior intestine (PI) tracts, characterized by morphological and functional differences. PS-MPs caused an increase of reactive oxygen species and nitrosylated proteins in both tracts, as well as augmented malondialdehyde production in the PI. PS-MPs also differently affected gene expression of antioxidant enzymes (i.e., superoxide dismutase, catalase, glutathione reductase). Moreover, an increased up-regulation of protective heat shock proteins (HSPs) (i.e., *hsp70* and *hsp90*) was observed in PI. Our findings demonstrate that PS-MPs are responsible for oxidative/nitrosative stress and alterations of detoxifying defense system responses with differences in AI and PI of gilthead seabreams.

## 1. Introduction

Microplastics (MPs) are plastic debris with a size in the range of 100 nm–5 mm, ubiquitous in aquatic ecosystems [1]. Their massive spread in recent decades is a significant problem because of the potential harmful effects induced on the environment and all living species, including humans. After introduction, mainly through the oral route, MPs can accumulate in the gastrointestinal tract (GI) or translocate through the blood and lymphatic circulation in body districts far from the introduction site [2]. Recently, MP particles have even been found in both the maternal and fetal portions of the human placenta, with possible adverse effects on the fetus that are not yet known [3].

In aquatic ecosystems, MPs tend to affect the environment by altering fauna and flora. The introduction of MPs during feeding results in their trophic transfer along the food chain, which in turn implies health problems for species placed at different steps of the chain [4].

In marine biota, MPs may induce a broad spectrum of adverse effects in experimentally controlled conditions, among which are obstructions throughout the digestive system, reduced growth performance, impaired reproduction, metabolic performance and immunotoxicity [5,6]. The extent of the effect is still disputed because of the diversity of possible combinations used among polymer type, size, shape, concentrations, time exposure and different experimental conditions [7,8]. MPs may also cause toxicity in marine organisms due to additives leaching out and the absorption of many harmful, persistent chemical contaminants, pharmaceuticals and microbial agents [9]. Cytotoxicity, inflammation, and oxidative stress (OS) are the most involved toxicity pathways [6,10]. MPs/NPs alone or in co-exposure with other pollutants induce OS, abnormalities of lipid metabolism and augmented mucus secretion in fish intestines [11,12,13].

An excessive intracellular production of reactive oxygen species (ROS) and reactive nitrogen species (RNS) is a well-recognized mechanism of toxicity induced by chemical pollutants [14]. ROS and RNS are mainly produced by cells of innate immunity, to quickly respond to stressing xenobiotics in teleost fish [15]. Neutrophils release ROS and RNS, allowing to counteract contaminants that have not been internalized because of their relatively large size [16]. Several in in vivo and in vitro studies evidenced the relationship between OS induction and MPs exposure following the immune response activation in the intestine [17,18].

Antioxidant enzymes neutralize low levels of ROS and RNS overproduction, however, when the extent of the latter is high, due to the prolonged and strong prooxidant stimulus, redox homeostasis fails. Therefore, a disruption of redox signaling and oxidative damage occurs [19]. The antioxidant enzymes are commonly analyzed as crucial biomarkers to evaluate the susceptibility of living organisms and fish exposed to chemical environmental pollutants [20,21].

MPs exposure may or not promote the response of antioxidant defenses, casting some doubt on their mechanism of action. Banaei and coauthors [22] observed both a decreased activity of glutathione reductase (GR), catalase (CAT) and glucose 6-phosphate dehydrogenase, and an increase of superoxide dismutase (SOD) and glutathione peroxidase (GPx) in the liver of common carp exposed to polyethylene (PE) for 30 days. Moreover, the decreased gene expression of SOD and CAT enzymes was observed in gut of marine medaka exposed to polystyrene (PS) [23].

This study aimed to evaluate the effect of the subchronic oral exposure (21 days) to PS-MPs particles on oxidative damage in the gut of gilthead seabreams (*Sparus aurata*). We examined the impact of PS-MPs on the production of ROS and malondialdehyde (MDA), and the expression of nitrosylated proteins, as the result of oxidative damage to lipids and proteins, respectively. The modulation of gene expression of some antioxidant defense system enzymes (SOD, CAT, GR) and heat shock proteins, HSPs (*hsp70*, *hsp90*), were also investigated as additional biomarkers of oxidative unbalance. Finally, histological analysis was performed to evaluate the intestine health status and the modifications of mucin layer.

In teleost fish, the anterior (AI) and the posterior intestine (PI) tracts are characterized by different physiological functions [24,25]. We focused on PS-MPs effects in AI and PI portions since little is known about the potentially different effects of MPs on the fish gut. The AI is mainly devoted to the absorption of fats and proteins [25], while the PI is primarily involved in triggering immune responses [24]. Gilthead seabream was chosen as experimental model for three main reasons: (i) it is, together with European seabass, widely the most diffused species in Mediterranean aquaculture fish [26]; (ii) it is a predatory species at the top of trophic chain frequently used as animal model and for toxicological and bioaccumulation studies; and (iii) it has ease of maintenance.

## 2. Materials and Methods

### 2.1. Dietary Preparation

Three different diets were formulated for the experiment: a standard diet consisting of pellets of commercial feed with a diameter of 4–4.5 mm, and two experimental diets composed of pellets of commercial feed enriched, respectively, with 0.5% and 5% of heterodisperse spherical PS-MPs particles in an odorless powder. PS particles had a concentration of 100% and a diameter between 1 and 20 μm (microparticles GmbH, Berlin, Germany). All diets were prepared by the same feed manufacturer (VRM S.r.l., NaturAlleva, Verona, Italy). Appendix A reports the ingredient composition of the standard diet.

### 2.2. In Vivo Experimental Procedures

Gilthead seabreams were obtained from a local fish farm (Soc. Coop. Acquamarina, Villa Literno, Caserta, Italy). Some 81 specimens were acclimated for 7 days into three tanks of 1000 L. Then, fish were weighed and randomly distributed into three groups divided into 6 fiberglass 250 L tanks (n = 27 animals for each group separated in two tanks).

The groups were defined as follows: (i) a control group (CON) fed with the standard diet; (ii) an experimental group fed with the diet containing the lowest PS-MPs concentration (PS1, 25 mg/kg b.w./day); and (iii) an experimental group fed with the diet containing the highest PS-MPs concentration (PS2, 250 mg/kg b.w./day). The treatment started after 14 days of acclimatization from the last weighing, and fish were fed twice a day following a rate of 1% body weight/tank/day. In particular, the CON group was fed with the standard diet both in the morning and in the evening, while the two experimental groups were fed with the diet containing PS-MPs in the morning, and with the standard diet in the evening. Administering the microplastic-enriched diet with the morning meal ensured that the animals ate the entire ration. Indeed, fish are famished after the night fast, guaranteeing proper animal exposure. The exposure lasted 21 days and at the end of the experimental time, fish were euthanatized by overexposure to an anesthetic (tricaine methanesulfonate-MS222, Sigma Aldrich, St. Louis, MO, USA). The AI and PI tracts were then excised and stored at −80 °C. With the label AI, we considered the intestinal tract just after the pyloric ceca; with the label PI, the intestine portion was just before the rectum.

### 2.3. Histological Evaluations

AI and PI were collected from three fish for each experimental group and rapidly placed in 10% neutral buffered formalin. Samples were processed, embedded in paraffin, sectioned at 5 μm, and stained with hematoxylin & eosin (H&E) for intestinal alterations.

To determine the modifications of mucin layer, goblet cells were identified and classified by Alcian Blue-Periodic Acid Schiff’s (AB-PAS) for neutral and acid (pH 2.5) mucin [27]. The five highest intact intestinal villi were randomly selected, and intestinal villi height was measured. The goblet cells from each section were counted using a high-power field (HPF; 400× magnification) and their number was calculated (goblet cell numbers/HPF) [28].

### 2.4. ROS Assay and MDA Measurement

ROS and MDA were measured as previously reported [29]. For ROS detection, an equal volume of freshly prepared homogenate of AI or PI was diluted in 100 mM potassium phosphate buffer (pH 7.4). It was added a final concentration of 5 μM dichloro-fluorescein diacetate (Sigma-Aldrich, Milan, Italy) in dimethyl sulfoxide for 15 min at 37 °C. The dye-loaded samples were centrifuged at 12,500× *g* per 10 min at 4 °C. The obtained pellet was mixed at ice-cold temperatures in 5 mL of 100 mM potassium phosphate buffer (pH 7.4) before being incubated for 60 min at 37 °C. The fluorescence was measured by the HTS-7000 Plus plate reader spectrofluorometer (Perkin Elmer, Wellesley, MA, USA) at 488 nm and 525 nm for excitation and emission wavelengths, respectively. ROS was quantified from the dichloro-fluorescein standard curve in dimethyl sulfoxide (0–1 mM).

For MDA measurement, AI and PI were homogenized in 1.15% KCl solution. A small amount of the homogenate (200 μL) was added to a reaction mixture containing 200 μL of 8.1% SDS, 1.5 mL of 20% acetic acid (pH 3.5), 1.5 mL of 0.8% thiobarbituric acid, and 600 μL of distilled water. The supernatant absorbance was spectrophotometrically measured at 550 nm, and the concentration of MDA was expressed as micromoles of MDA normalized on mg of protein of tissue homogenate (μM/mg protein). A standard curve was prepared using MDA bis (dimethyl acetal) as the source of MDA.

### 2.5. Real Time Semi-Quantitative-PCR Analysis

Total RNA was extracted from the AI and PI using TRIzol Reagent (Bio-Rad Laboratories, Hercules, CA, USA), following the instructions of RNA extraction kit (NucleoSpin^®^, MACHEREY-NAGEL GmbH & Co, Düren, Germany). cDNA was obtained using a High-Capacity cDNA Reverse Transcription Kit (Applied Bio-systems, Foster City, CA, USA) from 4 μg total RNA. All real time-PCR analyses were performed with a Bio-Rad CFX96 Connect Real-Time PCR System instrument and software (Bio-Rad Laboratories). The PCR conditions were as follows: 15 min at 95 °C followed by 40 cycles of two-step PCR denaturation at 94 °C for 15 s, annealing extension at 55–60 °C (depending on primer) for 30 s, and extension at 72 °C for 30 s. Each sample contained 500 ng cDNA in 2×QuantiTect SYBR Green PCR Master Mix and primers pairs (IDT Technologies, Coralville, IA, USA), to a final volume of 50 μL. Specific primers for target genes and housekeeping genes are listed in Appendix A.

### 2.6. Western Blot Analysis

Pieces of AI and PI of roughly 3 cm length were homogenized on ice in lysis buffer (20 mM Tris-HCl, pH 7.5, 10 mM NaF, 150 mM NaCl, 1% Nonidet P-40, 1 mM phenylmethylsulfonyl fluoride, 1 mM Na_3_VO_4_, leupeptin, and trypsin inhibitor 10 μg/mL). Cytosolic protein concentration was spectrophotometrically evaluated by using bovine serum albumin as standard, and 50 μg of proteins were subjected to SDS-PAGE. The blot was performed by transferring proteins from a slab gel to a nitrocellulose membrane using a Bio-Rad Transblot Turbo (Bio-Rad Laboratories, Hercules, CA, USA). The filter was then blocked at room temperature with milk buffer (1X PBS with pH 7.4, 5% non-fat dried milk and 1% NaF) for 45 min and probed at 4 °C overnight with anti-nitrotyrosine (Nox-Tyr, 1:1000, Merck Millipore, Billerica, MA, USA). Western Blot for anti-GAPDH (1:8000, Sigma-Aldrich, Milan, Italy) was performed to ensure equal sample loading. Bands were detected by ChemiDoc Imaging System (Bio-Rad Laboratories, Hercules, CA, USA).

### 2.7. Data and Statistical Analysis

Data are presented as mean ± standard error of the mean (S.E.M.). All experiments were analyzed using analysis of variance (ANOVA) for multiple comparisons, followed by Bonferroni’s post hoc test, using GraphPad Prism 9 (GraphPad Software, San Diego, CA, USA). Normality was tested using the Shapiro–Wilk test. To assess whether the variances across groups were homogeneous, the Brown–Forsythe’s test was used. Statistical significance was set at *p* < 0.05.

## 3. Results

### 3.1. Histological Evaluations

Animals fed with the control diet did not show significant pathologic findings in AI and PI by H&E staining (Figure 1 and Figure 2, panel a). Morphology assessment of the intestine from experimental groups (PS1 and PS2) revealed shortened intestinal folds that multifocally appeared blunt and necrotic (Figure 1 and Figure 2, panel b and c). Lamina propria and submucosa were diffusely expanded by a mild to moderate leukocyte infiltration. Intestinal villi height (by H&E) for AI and PI was measured in at least fifteen sections for each sample in CON, PS1 and PS2 groups. The mean was significantly lower in PS-MPs treated fish (Figure 1 and Figure 2, panel d).

The presence of goblet cells containing acid mucins was highlighted by AB-PAS double staining (Figure 1 and Figure 2, panel e, f and g). The mean number of intestinal goblet cells was significantly fewer in AI and PI of PS-MPs treated fish (Figure 1 and Figure 2, panel h).

### 3.2. PS-MPs Effect on ROS and MDA Production in AI and PI Portions

PS-MPs determined a dose-dependent increase of ROS production in the AI, which rose in significance at PS2 (Figure 3a). However, no differences were observed for MDA production (Figure 3b). Regarding the PI, a trend of augmented production of ROS and MDA was evidenced in a dose-dependent manner, with both being significant at the highest dose (Figure 3c,d).

### 3.3. PS-MPs’ Effect on Gene Expression of Antioxidant Enzymes and Nuclear Factor Erythroid 2-Related Factor 2 (Nrf2) in AI and PI Tracts

PS2 significantly increased gene expression of Cu/Zn SOD and decreased that of CAT in AI (Figure 4a,b), while PS1 did not modify both enzymes. No changes were also observed by PS1 and PS2 in gene expression of GR and Nrf2 in the same tract (Figure 4c,d).

In PI, SOD and GR did not change (Figure 5a,c), whereas the CAT and Nrf2 gene transcription were significantly reduced by PS2 (Figure 5b,d, respectively).

### 3.4. Effect of PS-MPs on Nitrosative Stress in AI and PI

As suggested by higher protein nitrosylation levels, PS-MPs induced increased nitrosative stress both in AI and PI. Such a result was significant at the highest dose in the AI and at the lowest dose in the PI (Figure 6a,b).

### 3.5. PS-MPs Effect on Gene Expression of HSPs in AI and PI Tracts

No changes were observed for HSPs (hsp70 and hsp90) gene expression at PS1 and PS2 in AI (Figure 7a,b). In the PI tract, the transcription of hsp70 gene increased at PS1, while that of hsp90 gene significantly increased at both doses (Figure 7c,d) without a dose-dependent correlation.

## 4. Discussion

Several and recent studies evidenced that MPs impact on the intestine of fish [30,31] inducing several alterations, among them dysbiosis, histological modifications, inflammation and OS [13,32,33].

The present study is the first to analyze oxidative and nitrosative damage determined by MPs in two intestinal tracts of aquatic species, namely the AI and the PI, characterized by different morphology and physiological functions.

OS is a key feature of MPs-induced toxicity and is studied mainly through the evaluation of ROS (superoxide anion, hydrogen peroxide, peroxynitrite, and other free radicals) and MDA production, and antioxidant defense system alteration [6]. Under normal conditions, ROS are produced in low amounts that are required to maintain cell homeostasis and signaling.

Environmental stressors can determine the overproduction of ROS and RNS, and the unbalance between the production of free radicals and the antioxidant defense system triggers and sustains OS and nitrosative stress. The main consequences of OS and nitrosative stress responses are the subsequent damage to lipids, proteins, RNA and DNA. Lipid peroxidation can disrupt the functions of cell membranes, thus compromising cell survival [34] and leading to the production of MDA, a biomarker of OS in living organisms exposed to environmental stressors [35]. Moreover, protein nitrosylation leads to the formation of stable derivatives (i.e., dityrosine) [36] not susceptible to enzymatic degradation, hence it is used as a biomarker of intracellular OS [37]. To counteract ROS and RNS overproduction, cells trigger signaling pathways, including the induction of a complex network of antioxidant and detoxifying enzymes (SOD, CAT, GPx, glutathione and HSPs) and transcription factors such as Nrf2 [38,39,40,41,42].

Here, oral exposure to PS-MPs caused increased ROS production in AI and PI tracts of gilthead seabream and an augmented MDA level in only PI, suggesting lipid peroxidation damage in this section.

Among the main antioxidant enzymes, SOD can convert superoxide radicals into hydrogen peroxide, then cleave into harmless water and oxygen by CAT. These enzymes are direct scavengers of free radicals, while GR inactivates secondary metabolites [43]. In our experimental conditions, we observed the increase in cytosolic SOD and the decrease in CAT in AI of PS2-treated fish, while GR and Nrf2 gene expression was unchanged. We may speculate that MDA production is counteracted by an enough efficient regulation of the SOD gene observed in AI, which limits ROS-induced oxidative damage to lipids, despite the reduced CAT gene expression. It could also be attributed to insufficient exposure time to determine lipid peroxidation in AI, as previously observed in the gut [44] and in other tissues [45] of *Sparus aurata.* The increase of oxidative damage biomarkers (MDA and protein carbonyl contents) and CAT activity was observed only after chronic exposure (90 days), whereas SOD activity increased already after 30 days, an exposure time similar to our experimental protocol [44]. Moreover, AI might be less responsive to the peroxidative damage than PI and thus need more time to evidence an MDA increase. This speculation is supported by the increase of ROS and MDA in PI observed in PS2 animals. More pronounced histopathological effects, mostly inflammatory alterations, were observed in distal intestine compared to the proximal one in goldfish exposed to virgin MPs [46], evidencing that in carnivorous teleost fish, PI is more susceptible and involved in the innate and adaptive immune response. Similarly, our data on histological analysis show an alteration in intestine health status after PS-MPs exposure. We observed that the height of intestinal villi and the number of goblet cells were lower in PS-MPs treated fish in both AI and PI. Goblet cells reside throughout the GI. They are responsible for producing and preserving a protective mucus blanket by synthesizing and secreting high molecular weight glycoproteins known as mucins. The mucus layer constitutes a dynamic protective barrier of GI tract, showing modifications in mucin production in response to intestinal injury [47].

Data from the literature indicate that the relationships between MPs exposure and oxidative injury in gut of aquatic species are complex, with results only sometimes in agreement. Increased ROS production and SOD activity were evidenced in the whole intestine of adult zebrafish subchronically exposed to MP fibers [48]. SOD activity also increased in the intestine of zebrafish exposed for few days to PE-MPs [49]; the same authors did not evidence an increase of MDA content. Other authors detected an increase of SOD and CAT activities and MDA production in the gut of zebrafish and rare minnow subchronically exposed to PS [40,50]. Hye-Min Kang and coauthors [23] observed an increased ROS production and reduced levels of both CAT and SOD enzymatic activities in the gut of marine medaka exposed for 14 days to PS-MPs. SOD and CAT decreased in Javanese Medaka fish exposed for 21 days with PS [51]. Interestingly, Zhang and coauthors [52] noticed an increased SOD activity in the intestine of silver carp acutely exposed to PS-MPs (5 μm, 80 μg/L in the water); however, MDA was also augmented, indicating the insufficient response of the antioxidant system.

Here, we examined the expression of Nrf2, a regulator of the physiological and pathological reaction to oxidant agents, that was decreased at the highest MPs dose in PI; no change was observed in AI, confirming the major susceptibility of PI to oxidative damage. Nrf2 directly regulates ROS homeostasis by controlling the antioxidant-mediated protective response through mechanisms that include the increased expression of signaling molecules and enzymes, including SOD [42].

The transcriptomic profile of several genes’ markers of antioxidant defense are expressed in both AI and PI of gilthead seabreams, and a significant increase for mitochondrial hsp70 was evidenced in AI [25]. HSPs, including HSP70 and HSP90, are highly conserved molecular chaperones involved in the folding and remodeling of proteins, and thus in their homeostasis and repair [53]. To the best of our knowledge, no study examined these proteins in the gut of gilthead seabreams. In our experimental conditions, a significant increase in both *hsp70* and *hsp90* gene expression was observed only in the PI. The lack of increase of the expression of these genes in AI might indicate that OS in this tract does not reach the level necessary to trigger a detectable response.

Several authors demonstrated that HSP proteins play a pivotal role in the response of aquatic biota to several stressors, among them ultraviolet radiation and chemical contaminants [39,54,55,56]. Interestingly, HSP70 expression was upregulated in the intestine of silver carp acutely exposed to PS-MPs [52] and in that of European sea bass, showing an increase or a decrease depending on the polymer type [32].

OS unbalance recorded in gilthead seabream intestine tracts may cause dysbiosis, barrier alteration, chronic inflammation, increased susceptibility to diseases and metabolic disorders, compromising the health status of fish exposed to MPs [30,57].

## 5. Conclusions

Our findings show that oxidative damage in gilthead seabream intestine is more evident in PI, but not always counteracted by antioxidant defense systems. Dissimilarities in response to PS-MPs exposure might depend on different susceptibility and responsiveness due to diverse physiological role of the intestine tracts. The results confirm that the gut is a target organ of MPs, since it is the first body district that ingested MPs encounter, affecting gastrointestinal health. The few conflicting literature results on this topic make it hard to compare data obtained in this study. However, it is conceivable that MPs-induced oxidative and nitrosative stress may damage organ function, leading to the alteration of fish health status and the onset of diseases.

## Figures and Tables

**Figure 1 animals-13-00606-f001:**
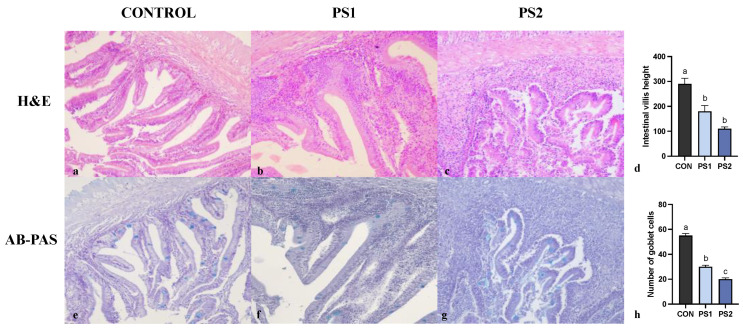
Representative microphotographs of anterior intestine (AI) of gilthead seabreams fed with control diet (**a**,**e**) and PS-MPs 25 mg/kg b.w./day (**b**,**f**) and 250 mg/kg b.w./day (**c**,**g**). Haematoxylin & eosin (H&E) staining is shown in panels (**a**–**c**) (original magnification, 20×). Alcian Blue-Periodic Acid Schiff’s (AB-PAS) staining is reported in panel (**e**–**g**) (original magnification, 20×). Panels (**d**,**h**) show the average of intestinal villi height (μm) and the number of goblet cells, respectively. All data are shown as mean ± S.E.M. Different letters indicate significant differences among treatments (at least *p* < 0.05).

**Figure 2 animals-13-00606-f002:**
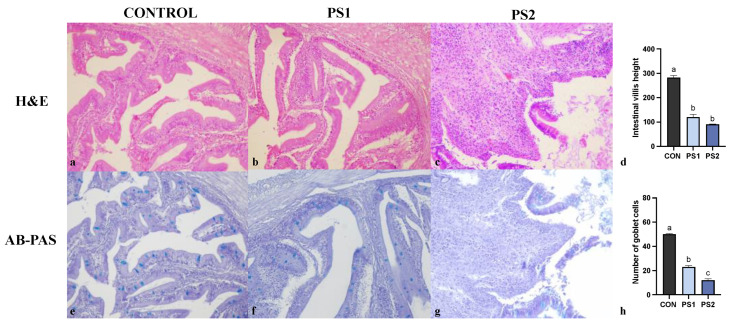
Representative microphotographs of posterior intestine (PI) of gilthead seabreams fed with control diet (**a**,**e**) and PS-MPs 25 mg/kg b.w./day (**b**,**f**) and 250 mg/kg b.w./day (**c**,**g**). Haematoxylin & eosin (H&E) staining is shown in panels (**a**–**c**) (original magnification, 20×). Alcian Blue-Periodic Acid Schiff’s (AB-PAS) staining is reported in panels (**e**–**g**) (original magnification, 20×). Panels (**d**,**h**) show the average of intestinal villi height (μm) and the number of goblet cells, respectively. All data are shown as mean ± S.E.M. Different letters indicate significant differences among treatments (at least *p* < 0.05).

**Figure 3 animals-13-00606-f003:**
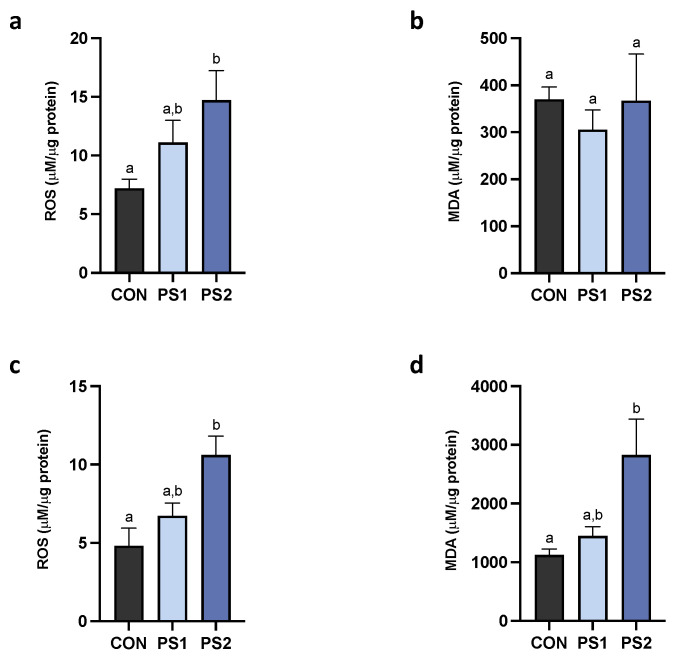
The oxidative stress induced by PS-MPs in the gut of gilthead seabreams. (**a**,**c**) ROS and (**b**,**d**) MDA levels were measured in the anterior (AI) and posterior intestine (PI), respectively. All data are shown as mean ± S.E.M. Different letters indicate significant differences among treatments (at least *p* < 0.05).

**Figure 4 animals-13-00606-f004:**
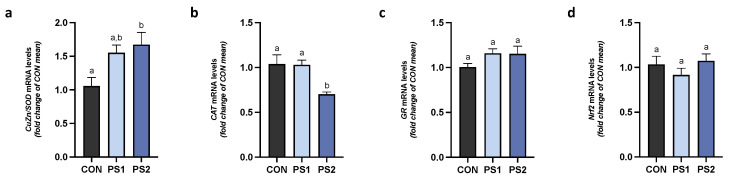
The effects of PS-MPs on antioxidant activity in the anterior intestine (AI). The transcriptional levels of (**a**) CuZn/SOD, (**b**) CAT, (**c**) GR, and (**d**) Nrf2 were measured after PS-MPs exposure in AI of gilthead seabream. All data are shown as mean ± S.E.M. Different letters indicate significant differences among treatments (at least *p* < 0.05).

**Figure 5 animals-13-00606-f005:**
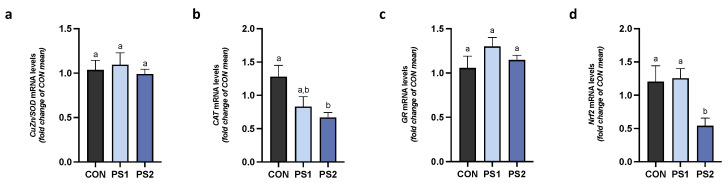
The alteration of antioxidant activity induced by PS-MPs in the posterior intestine (PI) of gilthead seabream. mRNA levels of (**a**) CuZn/SOD, (**b**) CAT, (**c**) GR, and (**d**) Nrf2 were measured after PS-MPs exposure in PI of gilthead seabream. All data are shown as mean ± S.E.M. Different letters indicate significant differences among treatments (at least *p* < 0.05).

**Figure 6 animals-13-00606-f006:**
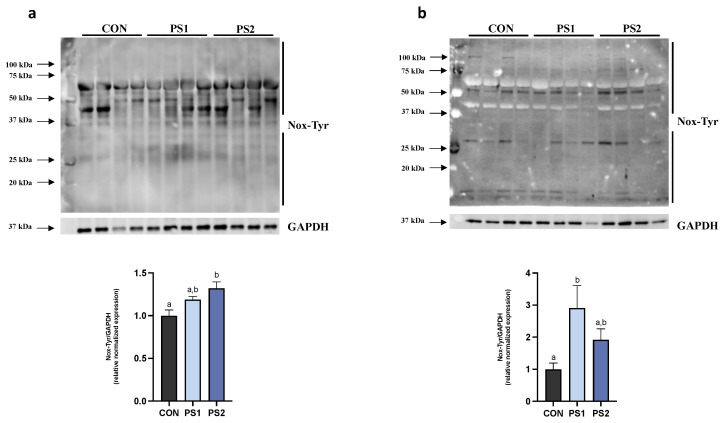
PS-MPs effect on nitrosylated proteins in anterior (AI) and posterior intestine (PI) of gilthead seabream. Western blot analysis for Nox-Tyr was performed in (**a**) AI and (**b**) PI of *Sparus aurata* after PS-MPs exposure. All data are shown as mean ± S.E.M. Different letters indicate significant differences among treatments (at least *p* < 0.05).

**Figure 7 animals-13-00606-f007:**
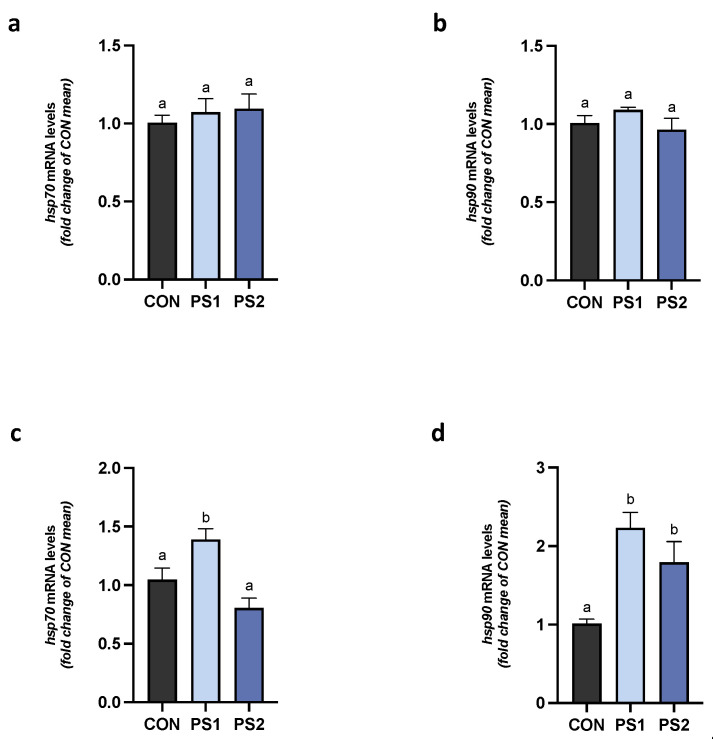
PS-MPs effect on gene expression of heat shock proteins in gut of gilthead seabream. The transcriptional levels of hsp70 and hsp90 were evaluated in (**a**,**b**) anterior (AI) and (**c**,**d**) posterior intestines (PI). All data are shown as mean ± S.E.M. Different letters indicate significant differences among treatments (at least *p* < 0.05).

## Data Availability

Data are available from the corresponding author on request.

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
