# Peer review of "Subchronic Exposure to Polystyrene Microplastic Differently Affects Redox Balance in the Anterior and Posterior Intestine of Sparus aurata"

_animals, 2023, doi:10.3390/ani13040606_

Round 1

Reviewer 1 Report

General comments:

The present work analyzed the expression changes of intestinal genes upon the exposure to microplastic in fish. The oxidative stress was revealed in gut and found with different response in the different segments. This is a worth trying to analyze the redox balance in the aquatic organism. However, the effect of microplastic exposure was not sufficiently revealed. There lacks histological evidence. At the molecular level, the tested few genes could just reflect the physical state very partially. Systemic studies, such as transcriptomic analysis is required. At least, the authors should consult more references, and analyze the genes related to more responsive pathways to the microplastic, even just focusing on the redox balance. Thus, a substantial revision is required.

Specific comments:

Introduction:

  1. Since a lot of studies focused on immune reaction, the relationship between redox balance and immune state should be introduced. More references should be included, such as "Immunotoxicity and intestinal effects of nano- and microplastics: a review of the literature" published in Particle and Fibre Toxicology.

  2. Other studies about the harmful effect of microplastic in fish species should be introduced, such as “Heterogeneity effects of nanoplastics and lead on zebrafish intestinal cells identified by single-cell sequencing” published in Chemosphere, and “Single-Cell RNA Sequencing Reveals Size-Dependent Effects of Polystyrene Microplastics on Immune and Secretory Cell Populations from Zebrafish Intestines” published in Environ. Sci. Technol. These systemic studies have revealed genes and pathways related to oxidative stress caused by microplastic exposure. Therefore, current experimental design could be reconsidered, accordingly.

Materials and Methods:

  1. Line 101: There should be a space either before or after the "=".

  2. Line 109: Why fed the standard diet in the evening?

  3. Line 166: "P" represented for the p value should be italic.

Results:

  1. There lacks the histological analysis. That's important to check the health state, in addition to the molecular findings. HE and Alcian blue staining could reflect the alteration of mucosa's structure and mucin layer respectively. See reference entitled "Oxidative Stress Causes Mucin Synthesis Via Transactivation of Epidermal Growth Factor Receptor: Role of Neutrophils" published in the Journal of Immunology.

  2. The color of the PS2's bar was too valid to distinguish.

  3. The control bands of GAPDH were not equal. Additional experiments are required. The target bands should be indicated by arrows.

Discussion:

  1. The structuring of the discussion lacks a clear logic.

Reviewer 2 Report

This study provides relevant and important information for the MP toxicology. However, I have some recommendations to make the MS more suitable for publication:

L31: Authors used division also with ^-1 and with the use of / symbol. I recommend using only one type consistently.

Introduction:

For an extensive summarize the enzymatic changes in fish due to exposure to chemicals related to MPs I can recommend the read and cite of the following manuscripts: https://www.sciencedirect.com/science/article/pii/S1532045622001818
https://www.mdpi.com/2305-6304/9/6/125

In this form, the introduction did not make the feeling for readers, that why and how MPs can affect aquatic ecosystems. 

E.g. a respiration rate test also can provide easy and good information about the stress status of investigated fish, see the firstly linked article. I can recommend this kind of investigation in your further research. 

L79-82: Authors should define precisely the anterior and posterior intestines, what are their borders in the organism for example. The MS have to be repeatable, and this is a key point. 

Material & methods: 

L91-96: Authors have to add the type of fish feeds properly. They gave the manufacturer, but also please describe the type precisely. 

Also please define much better the MPs. Chemical, physical type. It has to be repeatable. 

L98-112: Please add the number of replicates. In this case, there have to be at least 3 replicates per group. It means, for this study, a total of 9 aquaria have to be installed, 3 for each group and each aquarium has to contain the same number and condition of fish. Please also add statistical tests that provide the length and weight of fish in groups that were not statistically different.

 L163: Define "SEM" For ANOVA, normality and homogeneity of variances also have to be tested. For homogeneity of variances, I can recommend Levene's test. If data do not pass the test, authors have to use non-parametric tests, i.e., the Kruskal-Wallis test. 

Results

Fig 1-5: Bar diagrams can be suitable to interpret your data, but in this form, the quality is really poor. Please check Fig 5-7 in the above-mentioned references, e.g. here: https://www.sciencedirect.com/science/article/pii/S1532045622001818 

Also, the mark of significant differences with asterisk and line is really old-school and noninformative, because it is not transparent, which group differed and which is not. Therefore, I recommend using small letters. Please also check the above-mentioned references. 

Discussion

Why are one-line spaces between the paragraphs?

L212-215: The central part of MPs just goes through the digestive system, and they are not accumulating there. The accumulation can happen in the target tissues, i.e., muscle, liver, gills, bones, nervous system etc. 

I can recommend for further investigation that MPs will appear and accumulate in the above-mentioned target tissues or do they just go through the digestive system during this study?

Conclusions: 

I recommend also the rephrase of this part. Because as I mentioned above, MPs are not accumulating in the intestine. However, when they go through it, of course, they affect their health. 

Round 2

Reviewer 1 Report

The revision has improved a lot. While, for the histological data, there lacks quantitative analysis, such as the length of intestinal villi and the AB positive cells. The methods could be found in the referece entitled "The immunoregulatory role of fish specific type II SOCS via inhibiting metaflammation in the gut-liver axis" published in the journal  Water Biology and Security. In addition, for the hematoxylin's staining, the color was so light to show the nuclei. This is really important to show the immune cell infiltration in the LP layer. Additional experiment or adjusted color balance is reqired.

Reviewer 2 Report

The authors improved the quality of the MS, and they considered all reviewer's comments and suggestions. I feel that the MS is now suitable for publication in Animals. 

Author Response

We thank the reviewer for the suggestions that helped us to improve our research article